# Intranasal Delivery of *Quillaja brasiliensis* Saponin-Based Nanoadjuvants Improve Humoral Immune Response of Influenza Vaccine in Aged Mice

**DOI:** 10.3390/vaccines12080902

**Published:** 2024-08-09

**Authors:** Fernando Silveira, Florencia García, Gabriel García, José A. Chabalgoity, Silvina Rossi, Mariana Baz

**Affiliations:** 1Departamento de Desarrollo Biotecnológico, Instituto de Higiene, Facultad de Medicina, Universidad de la República, Av. Alfredo Navarro 3051, Montevideo 16100, Uruguay; florenciagarcia@higiene.edu.uy (F.G.); gabigarcia0898@hotmail.com (G.G.);; 2Departamento de Bioquímica Clínica, Instituto Polo Tecnológico, Facultad de Química, UdelaR, Ramal ‘‘José D’Elía” Ruta 101 y 8, Canelones 91000, Uruguay; srossi@fq.ed.uy; 3Department of Microbiology, Infectious Disease and Immunology, Faculty of Medicine, Université Laval, Quebec City, QC G1V 0A6, Canada; mariana.baz@crchudequebec.ulaval.ca

**Keywords:** *Quillaja brasiliensis*, adjuvant, nanoparticle, ISCOM matrices, IgA titers, influenza virus, delivered intranasally, viral challenge, aged mice

## Abstract

Increasing the effectiveness of vaccines against respiratory viruses is particularly relevant for the elderly, since they are prone to develop serious infections due to comorbidities and the senescence of the immune system. The addition of saponin-based adjuvants is an interesting strategy to increase the effectiveness of vaccines. We have previously shown that ISCOM matrices from *Q. brasiliensis* (IMXQB) are a safe and potent adjuvant. In this study, we evaluated the use of IMXQB as an adjuvant for the seasonal trivalent influenza vaccine (TIV) in an aged mice model. Herein, we show that subcutaneous injection of the adjuvanted vaccine promoted higher titers of IgM, IgG (and isotypes), and serum hemagglutination inhibition titers (HAI). Notably, aged mice immunized by intranasal route also produced higher IgG (and isotypes) and IgA titers up to 120 days after priming, as well as demonstrating an improvement in the HAI antibodies against the TIV. Further, experimental infected aged mice treated once with sera from adult naïve mice previously immunized with TIV-IMXQB subcutaneously successfully controlled the infection. Overall, TIV-IMXQB improved the immunogenicity compared to TIV by enhancing systemic and mucosal immunity in old mice conferring a faster recovery after the H1N1pdm09-like virus challenge. Thus, IMXQB nanoparticles may be a promising platform for next-generation viral vaccines.

## 1. Introduction

The influenza virus is responsible for causing severe illness in approximately 5 million individuals globally and causes death in the range of 290,000 to 650,000 individuals on an annual basis, as reported by the World Health Organization in 2023 [1]. Vaccination is considered to be the most effective intervention against influenza and its associated complications [2]. However, the effectiveness of influenza vaccines is limited even when they match the circulating strains in the population. Typically, current TIV and quadrivalent influenza vaccines (QIV) are not adjuvanted and only reduce disease incidence by 40–60%, and their efficacy is lower when the seasonal strains are not well-matched [3,4]. Moreover, the antibody response and protection provided by these vaccines tend to be lower among individuals aged 65 or older, as well as other vulnerable groups such as those with multiple health conditions or immunodeficiency [5]. Although global recommendations advise improving vaccination coverage among high-risk populations, suboptimal rates persist [4]. High-dose and adjuvanted inactivated influenza vaccines have been specifically developed to enhance immune responses in cohorts that have suboptimal immune responses to standard immunization such as, for instance, older adults who generally have low responses, mainly due to immunosenescence, comorbidities, and frailty [6]. The use of adjuvants in influenza vaccines has been shown to enhance antibody responses in older individuals, ultimately leading to improved protection against severe illness and a reduction in both morbidity and mortality [2,4,6]. The ones that have been approved are alum, MF59, AS03, AF03, virosome, and Matrix-M^TM^ [6,7,8], and the development of next-generation influenza vaccines is underway.

Alternative adjuvants, specifically nanoadjuvants, have been incorporated into modern vaccines formulation as a strategy to counteract the immune system’s reduced responsiveness in older individuals as demonstrated by recent research [9].

Saponin-based adjuvants (SBA) from the *Quillaja* genus are emerging as an alternative to traditional adjuvants and currently used in several licensed vaccines, as they stimulates effective Th1-biased immune responses to combat intracellular pathogens [10,11,12,13]. Recently, Novavax, Inc. has developed a COVID-19 vaccine called NVX-CoV2373 based on recombinant spike protein of the SARS-CoV-2 virus with Matrix-M^TM^, a saponin-based nanoparticle adjuvant. Clinical trial have shown that this vaccine is safe, provides durable protection against the virus [12,14,15], and has less reactogenicity compared to mRNA vaccines [16]. In addition, the malarial vaccine Mosquirix™ [17,18] and varicella zoster virus (RZV) Shingrix^®^ vaccines [11,19], developed by GSK, also contain saponins as an adjuvant (AS01) [20,21]. In particular, Shingrix^®^ is an approved vaccine in several countries for adults aged 50 and older. Its protective efficacy has been shown to wane minimally over four years, and sustained, in individuals aged 70 and older [5,19].

Recently, using an adult mice model, we demonstrated that delivering a TIV with *Q. brasiliensis* saponin-based nanoadjuvants, such as ISCOMs (IQB90) [22] or ISCOM matrices (IMXQB) [23], can enhance the effectiveness of the vaccine. These nanoparticles induced high levels of IgG2a and IgG1 antibodies, which have virus-neutralizing capabilities and have also improved serum hemagglutination inhibition titers. After the challenge with A/Uruguay/897/2018 (H1N1)pdm09 virus, it was observed that the animals receiving TIV adjuvanted-IMXQB (TIV-IMXQB) had significantly lower viral titers in their lungs compared to those vaccinated with TIV alone. Importantly, mice that received the TIV-IMXQB vaccine by intranasal delivery and challenged with a lethal dose of influenza virus had no weight loss or mortality, and showed full protection against lung virus replication. In contrast, animals vaccinated with TIV alone experienced a high mortality rate [23].

In this study, we compared the immunogenicity of the TIV-IMXQB and TIV alone in mice aged 14–15 months old. Our results indicate that the TIV-IMXQB experimental vaccine elicits a stronger immune response than the TIV vaccine when administered either subcutaneously or intranasally, and this response is effective to provide protection against influenza virus infection.

## 2. Material and Methods

### 2.1. ISCOM Matrices Adjuvant: Preparation and Characterization

*Quillaja brasiliensis* (A. St.-Hil. et Tul) Mart. leaves were collected at Parque Battle, Montevideo, Uruguay (–34.89302, –56.15727). Voucher MVFQ 4321 was deposited at the Herbarium of the Facultad de Química, Universidad de la República. The extraction and purification of *Quillaja brasiliensis* saponin fractions (QB) [24] and the ISCOM matrices (IMXQB) [25] were carried out as previously described. The IMXQB nanoadjuvants obtained were sterilized by filtration using a 0.22 μm syringe filter and maintained at 4 °C until use.

In previous reports, we demonstrated that IMXQB nanoadjuvants are safe regarding cell viability and hemolytic activity without evidences of local toxicity (local swelling, loss of hair, and piloerection) when administered in mice [22,23,26,27,28].

### 2.2. Experimental Vaccines Formulations and Mice Immunization

Sanofi Pasteur’s commercial trivalent inactivated influenza virus vaccine (TIV, VAXIGRIP, year 2021), involving influenza virus strains A/Victoria/2570/2019 (H1N1)pdm09, A/HongKong/2671/2019(H3N2), and B/Washington/02/2019 (15 µg of hemagglutinin (HA) from each strain per 0.5 mL) was used as immunogen to formulate the IMXQB-adjuvanted vaccine (TIV-IMXQB).

TIV-IMXQB was prepared mixing required volumes of commercial TIV, IMXQB, and saline solutions in order to obtain 2.5 µg of each HA (TIV) and 5.0 ug IMXQB in 100 µL/dose. IMXQB amount was defined as the saponin content in the nanoparticle adjuvant. For non-adjuvanted TIV vaccine, required volumes of commercial TIV and saline solutions were mixed in order to achieved 2.5 µg of each HA (TIV) in 100 µL/dose. Experimental vaccines were prepared under aseptic conditions, filtered through 0.22 µm, and kept at 4 °C until administration.

The protocol of this study, Exp. Nº 070151-000006-23, was ethically approved by “Comisión Honoraria de Experimentación Animal” (CHEA-Universidad de la República, Uruguay). Animals were appropriately housed in microisolators with filter tops in a biosafety level 2 (BSL-2) laboratory at a controlled temperature (22 ± 2 °C) and air humidity (50–60%) with a 12/12 h light/dark cycle and with access to food and water ad libitum.

Female BALB/c mice were purchased from Dirección de Laboratorios Veterinarios (Ministerio General de Agricultura y Pesca, Uruguay) and aged for until 14/15 month old at the Instituto de Higiene (Universidad de la República). Female CD1 mice (two-month-old) were produced at the Instituto de Higiene (Universidad de la República).

Aged female BALB/c mice were divided into 6 groups (*n* = 19 per group) and immunized twice (on day 0 and 14) subcutaneously or intranasally, with TIV-IMXQB, TIV, or saline solutions, respectively. For subcutaneous administration (s.c.), animals were injected in the hind neck with 100 µL of the corresponding vaccine formulation (total 7.5 µg/dose) or saline solutions. For intranasal delivery (i.n.), mice were anesthetized with ketamine–xylazine and received half the dose used for s.c. immunization, i.e., 3.75 µg/dose of TIV of total HA in 50 µL of the corresponding vaccine formulation or saline solution. Blood samples were collected by retro-orbital venous plexus puncture, for all mice, on days 0, 14, and 28, prior to immunization, and day 120, prior to challenge. Sera were kept at −20 °C until they were used.

### 2.3. Analysis of Antibodies Responses

Anti-TIV IgM was evaluated at day 0 and 14, and IgA, IgG, and anti-isotypes (IgG1 and IgG2a) classes were measure at 0, 14, 28, and 120 days post-priming (dpp) by indirect ELISA method as previously described [22,23]. Class-specific secondary antibodies labelled with HRP were purchased to Southern Biotech, Birmingham, AL, USA, and used appropriate dilution. For IgG and isotypes determination, sera were appropriately diluted and antibody titers were expressed as arbitrary units per mL (AU/mL) using a standard curve built with a pool of sera from immunized groups. For IgM and IgA determination, sera were diluted 1/500 and titers were expressed as optical density (OD) because signal measures were much lower than for IgG and isotypes.

### 2.4. Avidity Index for IgG and IgA Elicited by Vaccines Formulations

Serum IgG avidity was determined by 3 M urea denaturation, as described elsewhere for us [28]. Briefly, ELISA plates (Greiner Bio-One, Darmstadt, Germany) were coated overnight at 4 °C with the TIV (1.0 µg/mL) diluted in PBS (pH 7.2). Plates were then washed three times with PBS containing 0.05% Tween^®^ 20 (PBS-T20) and blocked with 1% Tween^®^ 20 in PBS at 37 °C for 2 h. Appropriately diluted serum samples were added in quadruplicate (100 µL/well) and incubated for 1 h at 37 °C. After that, plates were washed three times with PBS-T20 and 100 µL of 3 M urea was added to one half of the wells and PBS-T20 (100 µL) was added to the other half of the plate. After 30 min incubated at 37 °C. Plates were washed as before. HRP-labelled secondary antibody (anti-mouse IgG or anti-mouse IgA, Southern Biotech, Birmingham, AL, USA) were incubated 1 h at 37 °C and washed three times with PBST before adding the substrate solution (TMB). The procedure was completed by stopping the reaction with 30 µL/well of 1N H_2_SO_4_ and measuring OD at 450 nm in an ELISA plate reader (Sunrise). The avidity index (AI) was calculated using the following formula: (mean Abs of urea-treated wells/mean OD urea-untreated wells) × 100% [28].

For each serum, proper dilutions were considered when antibodies did not saturate completely the immobilized antigen and the OD readings lay between 0.6 to 0.9 absorbance units. Serum samples that did not comply with these requisites were not analyzed. For subcutaneous immunization, IgG avidity index was determined using 1/3000 dilutions for serum samples from the TIV-IMXQB group, while 1/500 dilutions were used for serum samples from the TIV group. Similarly, for i.n., IgG avidity index was determined using 1/1000 dilutions for serum samples from the TIV-IMXQB group, while 1/100 dilutions were used for serum samples from the TIV group. For i.n., IgA avidity index was determined just for the TIV-IMXQB group using 1/100 dilutions, as specific-serum IgA could not be detected for the TIV group.

### 2.5. Hemagglutination Inhibition and Microneutralization Assay

Antibody titers against influenza A/Victoria/2570/2019 (H1N1)pdm09-like virus in mouse serum samples were measured by a hemagglutination inhibition (HAI) assay as previously reported [23]. Briefly, for HAI, non-specific inhibitors were removed from mice sera by overnight treatment with receptor-destroying enzymes (Denka Seiken, Tokyo, Japan). Sera were twofold serially diluted in 96-well plates starting at a dilution of 1:10, and 4 HA units of virus were added. Control wells received phosphate-buffered saline (PBS) alone. After 30 min of incubation at room temperature, 50 µL of a 0.5% guinea pig red blood cell solution was added to the mixture, and then the mixture was incubated for 45 min before evaluation of hemagglutination. The HAI titer was recorded as the reciprocal of the highest dilution of serum at which hemagglutination was inhibited.

For the MN assay, sera were first inactivated at 56 °C, and serial twofold dilutions were prepared, starting at 1:20 dilution. Equal volumes of serum and A/California/07/2009 (H1N1)pdm09-like virus were mixed and incubated for 60 min at room temperature. The residual infectivity of the virus–serum mixture was determined in MDCK cells using four wells for each serum dilution. Neutralizing antibody titer was defined as the reciprocal of the serum dilution that completely neutralized the infectivity of100 TCID50 (median tissue culture infectious dose) of A/California/07/2009 (H1N1)pdm09-like as determined by the absence of a cytopathic effect (CPE) on MDCK cells at day 4 and calculated using the Reed–Muench method [29].

### 2.6. Mouse Challenge

The influenza virus used for the challenge (A/Uruguay/897/2018 (H1N1)pdm09-like virus) was isolated from a nasal discharge sample of a female patient in Montevideo, Uruguay. The isolated was propagated in MDCK using standards techniques and a viral stock was stored at −80 °C until use.

Three months after the first immunization, BALB/c mice of 17 or 18 months of age were challenged with A/Uruguay/897/2018 (H1N1)pdm09-like virus. Animals were anesthetized with ketamine–xylazine anesthesia and 50 µL saline solution containing 1 × 10^6^ TCID_50_ virus particles was administered intranasally, to each one. Mice (n = 5) were observed for 14 days post-infection and monitored for weight loss and other clinical signs of virus-induced morbidity daily (ruffled fur and lethargy); mice were euthanized if weight loss exceeded 20% of initial body weight.

### 2.7. Passive Antibody Administration: Mice and Infection

Female CD1mice, 6 to 8 weeks old were divided into three groups (n = 5/group), named TIV, TIV-IMXQB, and IMXQB, respectively. Immunization was performed on days 0 and 14 by s.c. route, in the hind neck with 100 µL of TIV in saline solution (7.5 µg total HAs/dose), TIV-IMXQB (7.5 µg total HAs + 5.0 µg saponin/dose), or IMXQB (5.0 µg saponin /dose), according to each group. Bleedings were performed on day 28, 14 days after second immunization, and obtained sera were stored at −20 °C until use.

Sera from each group were pooled, evaluated for neutralizing activity by microneutralization assay and HAI titer, and administered to experimentally infected 15-month-old BALB/c mice for passive immunization. The TIV-immunized group had a HAI and neutralizing antibody titer of 1280 and 905, respectively, while the TIV-IMXQB-immunized group had 2560 and 7241, respectively, whereas the control (IMXQB-immunized group) had 160 and 40, respectively [23]

Aged female BALB/c mice (15-months-old) were infected intranasally, with 1 × 10^4^ TCID_50_ of A/Uruguay/897/2018 (H1N1)pdm09-like virus in saline solution (50 µL/mouse), previously anesthetized (n = 22). Twenty-four hours later, mice were divided into three groups and received pooled sera from TIV-immunized CD1 mice (n = 7), TIV-IMXQB-immunized CD1 mice (n = 7), IMXQB-immunized CD1 mice (n = 5), or saline solution (n = 3). Each mouse received 100 μL of the corresponding sera diluted in 100 μL saline solution or 200 μL of saline solution (control group), intraperitoneally. Mice were observed for 21 days post-infection and monitored for weight loss and other clinical signs of virus-induced morbidity daily, as described in Section 2.6.

### 2.8. Statistical Analysis

Statistical significance was assessed by one-way-ANOVA Kruskal–Wallis test with uncorrected Dunn’s post-test correction for multiple comparisons compared to the control group (unadjuvanted TIV antigen). The Mann–Whitney test to compare two groups was also used. GraphPad Prism version 10.01 (GraphPad Software, Inc., Boston, MA, USA) was used for data analysis.

## 3. Results

### 3.1. Subcutaneous Immunization with TIV-IMXQB Significantly Improved Anti-TIV Humoral Immunity and Protected Aged Mice against Lethal Challenge with A/Uruguay/897/2018 (H1N1)pdm09-like Virus

A commercial vaccine TIV, either alone or adjuvanted with ISCOM matrices *Q. brasiliensis* saponin-based (TIV-IMXQB), was administered subcutaneously to 14–15 month old mice; a control group received saline via the same route. The experimental design is schematized in Figure 1. ELISA evaluation of anti-TIV IgM antibodies did not show significant differences among groups vaccinated with either TIV-IMXQB or TIV on day 14 (Figure 2A). Likewise, specific anti-TIV IgG antibodies were determined by ELISA on days 14, 28, and 120 dpp. The IgG antibodies specific to TIV were significantly increased in the TIV-IMXQB group on day 14 and 28 after priming, compared to the unadjuvanted vaccine (Figure 2B, *p* = 0.011 and *p* = 0.014, respectively). As expected, the immune response in the vaccinated groups was boosted after the second-shot immunization; however, on the day 120 after the first shot, both groups showed similar behavior and antibody levels dropped substantially to priming levels (Figure 2B).

In regard to the IgG isotype (IgG1 or IgG2a) levels, no differences were observed when comparing TIV-IMXQB and the unadjuvanted vaccine (Figure 2C,D), with the exception of the specific IgG1 antibodies on day 28 post-priming, where mice vaccinated with TIV-IMXQB (*p* = 0.03) showed a significant increase compared with the unadjuvanted vaccine (Figure 2C).

In order to better understand the humoral response elicited by the nanoadjuvants formulation against the A/H1N1 influenza virus, we assessed the avidity of IgG responses and conduced hemagglutination inhibition assays. Figure 2E summarizes the avidity of IgG antibodies induced by TIV-IMXQB or TIV vaccine formulations. In all immunized mice except in one animal from each group, the avidity index determined on day 28 post-priming was greater than 60% with no significant differences between groups and stayed quite constant at day 120 post-priming. Avidity was not evaluated for animal sera with low absorbance signals in ELISA test. However, it is worth mentioning that due to differences observed for specific IgG levels, appropriate serum dilutions used to avidity evaluation were different for TIV-IMXQB and TIV groups (1/3000 and 1/500, respectively).

HAI geometric mean titers (GMT) on days 28 and 120 post-priming did not show significant differences between the TIV-IMXQB (GMT of 80 and 53, respectively) and TIV alone (GMT of 85 and 17, respectively), although TIV-IMXQB showed a tendency to higher HAI titers than TIV alone on day 120 (GMT value 53 vs. 17 respectively) (Figure 3A).

One hundred twenty days after the first immunization (18-month-old mice), animals were intranasally challenged with a lethal dose (1 × 10^6^ TCID_50_/50 µL) of A/Uruguay/897/2018 (H1N1)pdm09-like virus and monitored daily for clinical signs of disease, weight loss, and mortality for 14 days. Significant weight loss was observed in non-immunized mice (saline group) starting around day 1 post-infection and 60% of them experienced rapid deterioration and died by day 7 (Figure 3B,C). Although all immunized animals survived virus challenge, mice receiving TIV experienced the highest loss of their body weight (10%) between day 3 to 5 and then recovered, while no weight loss or other signs of disease were evident along the 14 days of the experiment in mice receiving TIV-IMXQB (Figure 3B,C).

### 3.2. Intranasal Immunization with TIV-IMXQB Induces Higher Antibody Levels than the Commercial Unadjuvanted Vaccine and Promotes a Rapid Recovery against Lethal Challenge

Groups of mice were immunized intranasally with TIV, TIV-IMXQB, or received only saline (control group) (Figure 1). Serum IgM and IgG antibody responses for i.n. IMXQB and commercial TIV are depicted in Figure 4. At 14 dpp, significant differences were observed for IgM (*p* = 0.003) and IgG (*p* = 0.017) levels between groups (Figure 4A,B, respectively). After second administration, all animals in the TIV-IMXQB group had significantly high levels of serum IgG against TIV antigens (28 dpp; Figure 4B, *p* = 0.0002) and a less pronounced decline at the end (120 dpp; Figure 4B, *p* = 0.052) compared to s.c.. In contrast, serum IgG antibody responses elicited by commercial TIV vaccine were weaker compared to the TIV-IMXQB vaccine at all time points (14, 28, and 120 dpp, Figure 4B), as suggested by measured IgG levels for both groups, with serum IgG undetectable for some mice from TIV group.

Similar observations arose from IgG subclasses (IgG1 and IgG2) analysis. Twenty-eight days after priming, the TIV-IMXQB immunized group showed significant higher levels of IgG1 (Figure 4C) and IgG2a (Figure 4D) than the TIV-immunized group (*p* = 0.001 and *p* = 0.037, respectively). At 120 dpp, differences were observed between the TIV and TIV-IMXQB groups. On this day, the median IgG1 value was 7 times higher for TIV-IMXQB compared to TIV, while the median IgG2a value was 11 times higher for TIV-IMXQB compared to TIV. However, the Kruskal–Wallis test could not establish significant statistical differences. (Figure 4C,D).

Avidity indexes of IgG responses due to TIV-IMXQB or TIV vaccine formulations are shown in Figure 4E. Both vaccine formulations promoted IgG responses characterized by avidity indexes over 60% in most animals on days 28 and 120 after priming, without significant differences between them (Figure 4E).

Serum IgA against TIV antigens were detected only in mouse sera from the TIV-IMXQB group on days 28 and 120 post-priming (Figure 5A; *p* = 0.001 and *p* = 0.007, respectively), with low to moderate avidity indexes, (below 50%) as shown in Figure 5B. Serum IgA against TIV antigens was undetectable in serum samples from mice immunized with commercial TIV vaccine. Thus, IgA’s avidity determination could not be performed in this group.

HAI assays also showed an improved antibody response for TIV-IMXQB-immunized group compared to the commercial TIV-immunized one. Significant differences were observed at 28 (*p* = 0.0006) and 120 (*p* = 0.039) dpp, with higher HAI titers for the TIV-IMXQB group (98 vs. 32 and 80 vs. 7, respectively) (Figure 6A).

One hundred twenty days after the first immunization, all mice were intranasally challenged with a lethal dose (1 × 10^6^ TCID_50_) of A/Uruguay/897/2018 (H1N1)pdm09-like virus and monitored daily for clinical signs of disease, weight loss, and mortality for a 14 days period. All mice (immunized or not) experienced a 10% weight loss from day 3 to day 5 after experimental infection. From day 6, mice immunized with TIV-IMXQB vaccine begun to gain weight and experienced a fast recovery. Mice immunized with commercial TIV vaccine, begun to recover later, on day 9. Mice from non-immunized group experienced rapid deterioration from day 4 and died between days 7 to 11, with a 20% survive at the end of the experiment (Figure 6B,C).

### 3.3. Passive Transfer of Sera Obtained from Adult Mice Immunized with TIV-IMXQB Promote a Faster Recovery of Old Mice after Virus Challenge

Fifteen-month-old mice were intranasally challenged with a sublethal dose (1 × 10^4^ TCID_50_/50 µL) of A/Uruguay/897/2018 (H1N1)pdm09-like virus (see Appendix A). Twenty-four hours after the viral challenge, passive immunizations (intraperitoneal injection) were performed with pooled sera from adult (about 3-months-old) mice previously immunized subcutaneously with commercial TIV, TIV-IMXQB, IMXQB (only adjuvant), or saline solution. Mice were monitored daily for clinical signs of disease, weight loss, and mortality for 21 days.

Passive transfer of pooled sera from animals vaccinated with TIV-IMXQB (higher HAI and MN antibody titer) to the aged mice resulted in minor weight loss (5% of their initial body weight) and underwent a significantly faster recovery from infection compared to those treated with pooled sera from animals vaccinated with TIV, IMXQB, or saline solution (Figure 7; *p* = 0.0001, 0.0097, or 0.0036, respectively). By the end of the study (day 21 post-viral infection) only those animals that received the TIV-IMXQB pooled sera returned to baseline. Furthermore, no significant differences were observed for weight recovery between the aged mice that received sera from adult mice immunized with TIV or inoculated with IMXQB or saline at the end of the study (Figure 7).

## 4. Discussion

Vaccination against influenza virus is a recommended practice every autumn to prevent severe respiratory disease and commercial vaccines have existed since the 1940s. Currently, three types of licensed seasonal influenza vaccines are available: inactivated, live attenuated, and recombinant HA vaccines [7]. The TIV and QIV are not adjuvanted and confer limited protection, especially in young children and the elderly [6,7]. Consequently, the necessity of more efficient vaccines is still a matter of concern, especially for the elderly (>70 years-old), where several attempts have been made to increase influenza vaccine efficiency [5,6,30] such as dose increase, inclusion of adjuvants, and change in route of administration [6,30].

It is well-known that vaccines must stimulate both innate and adaptive pathways in order to confer long-lasting protection. For non-living vaccines, multiple administrations and the incorporation of adjuvants in vaccine formulations work in favor to this aim. Adjuvants enhance the immune response to vaccines, increasing both the potency and quality of adaptive immune response, providing the highest level of protection. Additionally, adjuvants may help to reduce the vaccine dose and the number of doses required to reach protecting immunity [3,6,21].

Quillaja genus saponin-based delivery systems outperform conventional vaccines, promoting lymph node accumulation and antigen aggregation, as well as antigen uptake and preservation from catabolism, enhancing the overall immune response [10,15,21]. The commercial *Q. saponaria* saponin-based Matrix-M^TM^ adjuvant is safe [8,10,14,15,31,32]. Similarly, IMXQB cage-like structures (diameter average 40 nm in size), obtained by mixing *Q. brasiliensis* saponins and sterols [25], have been shown to be non-toxic in several studies [22,23,27]. The same attribute of safety, i.e., no signs of local toxicity such as local swelling, loss of hair, and piloerection, were observed in this study, where aged mice were inoculated with the IMXQB-formulated vaccine.

We have previously demonstrated that the combination of a commercial TIV adjuvanted with our IMXQB nanoparticles improved the adaptive immune response of commercial TIV alone and conferred superior protection against infection in adult mice [22,23]. In the present study, we formulated an experimental vaccine consisting on the commercial TIV adjuvanted with IMXQB nanoparticles and we tested it in an aged mouse model. TIV-IMXQB effectiveness, humoral immune response, and protection against experimental infection, were compared to that of the same unadjuvanted commercial TIV. The incorporation of IMXQB nanoparticles to commercial TIV strongly promoted the generation of specific IgG in more than 50% of the mice (12/19) as early as 14 dpp, compared to the unadjuvanted commercial vaccine. Two weeks after the second shot, a boosting effect was observed in both vaccinated groups, with a higher increase in IgG levels for the TIV-IMXQB group (*p* = 0.014). Moreover, TIV-IMXQB promoted a sharp increase in the levels of IgG1 compared to commercial TIV (14 days post booster; *p* = 0.03), which waned towards 120 dpp. We also detected IgG2a antibodies in sera from mice immunized with both vaccines (TIV and TIV-IMXQB) on day 28 dpp, but significant differences were not observed between vaccinated groups. At 120 dpp, each vaccinated group had five mice; as shown in Figure 2D, three of five mice from the TIV-IMXQB group kept IgG2a levels comparable to those observed by 28 dpp, contrary to the drop observed for mice from the TIV group. The profiles of IgG2a and IgG1 isotype induction are in accordance with our previous reports with adult mice in both experimental Zika virus [27,28] and influenza vaccines [22,23].

The avidity of IgG responses showed no differences between both vaccinated groups. All but one of the serum samples analyzed from each vaccinated group showed high avidity (avidity indexes > 60%), while just one mouse in each group showed moderate avidity (avidity indexes around 40%). High avidity IgG responses continued, at least, until 120 dpp, as expected after two doses of vaccine.

HAI assays, the gold standard method for measuring the functional antibody response following influenza virus vaccination, showed that both vaccines successfully induced high titers of antibodies capable of inhibiting viral hemagglutinin. Moreover, at 120 dpp, inhibitory functionality was still present in all mice from the TIV-IMXQB group.

Notably, after 120 days from the first vaccination (18/19-month-old mice by then), animals inoculated with TIV and challenged with a lethal dose of A/Uruguay/897/2018 (H1N1)pdm09-like virus lost >10% of their body weight by day three, and recovered by day six. However, the animals that received the TIV-IMXQB-adjuvanted immunization did not exhibit the same body weight loss as the animals in the control group did, indicating a faster recovery from the virus. During a 14 day period since infection, weight loss was greater in both the TIV and control groups. Deaths were only observed in the control group (60% of the mice).

Other research groups also followed the strategy of incorporating adjuvants and tested the adjuvanted formulations in aged-mice models, with the aim to improve the protection conferred by inactivated influenza vaccine. Vassilieva et al. [33,34] found that the combination of cGAMP and Quil A^®^ induced a more effective protection in Balb/c aged mice, administered by intradermal or intramuscular routes; however, the improvement observed was not as potent as that observed in adult mice. An opposite effect was reported by Ramirez et al., when Fluzone^®^ vaccine adjuvanted with CpG was inoculated by intramuscular route in C56BL/6J aged-mice; in this study CpG-adjuvanted vaccine induced complete protection to adult mice but failed in aged mice [35]. Both adjuvants investigated in aged mice, cGAMP+Quil A^®^ and CpG, resulted in elevated IgG titers and low HAI titers, but differed in survival rate virus challenge. cGAMP+Quil A^®^-adjuvanted vaccine prevented virus infection’s mortality and morbidity, while CpG did not. Our data show that aged mice vaccinated with TIV-IMXQB (s.c.) also experienced a lower response compared to adult mice receiving the IMXQB-adjuvanted seasonal influenza vaccine and same vaccination schedule [22,23], but in both cases, adult and aged mice, elevated HAI titers were observed as well as reduced morbidity to the virus challenge.

The most efficient way to combat respiratory infections disease is to provide immunization via mucosal surfaces (eliciting antibodies and robust T-cell immunity). This can be accomplished by designing safe vaccines capable of stimulating both systemic and mucosal immune pathways, providing long-lasting immunity. Furthermore, adequate stimulation of the immune system through the mucosa of the nasopharyngeal tract leads to systemic and mucosal immunity, controlling the pathogen at the site of entry [36,37]. According to this study, the intranasal delivery of TIV-IMXQB effectively triggered a humoral immune response with high levels of serum antibodies, IgG, IgG1, IgG2a, and IgA, after the second shot, which remained by at least 120 dpp. Indeed, intranasal delivery of commercial TIV vaccine failed to stimulate detectable systemic antibody responses, while this was achieved by TIV-IMXQB. It is worth mentioning that the commercial TIV used in this study was licensed for intramuscular or deep subcutaneous route administration. However, the incorporation of IMXQB nanoparticle adjuvant allowed the stimulation of the immune system when administered by intranasal route, as demonstrated by the serum antibody responses measured, suggesting that the stimulation of mucosal immune system occurred. Comparing antibody systemic responses elicited by TIV-IMXQB administrated by both subcutaneous or intranasal routes, the kinetics of antibody response decayed a slightly faster when administered subcutaneously. As suggested by the IgG level on day 120 pp, the intranasal route led to superior IgG levels, even though the dose was half of that used subcutaneously. Regarding IgG1 and IgG2a subclasses, the intranasally delivered TIV-IMXQB stimulated both isotypes in comparison to TIV, as previously reported [22,23]. At 28 dpp, IgG1 and IgG2a levels in the TIV-IMXQB group showed significant differences compared to the TIV group (*p* = 0.001 and *p* = 0.037, respectively). In addition, differences between levels of IgG1 and IgG2a were observed; intranasal vaccination resulted in higher levels of IgG1 and lower levels of IgG2a, compared to subcutaneous administration.

Serum IgA responses were evaluated just for intranasal delivery vaccines, assuming that they would be favored by this route. IgA against TIV antigens was detected in sera from mice belonging to the intranasally administered TIV-IMXQB group and had moderate avidity compared to the IgG response. However, IgA levels waned by 120 dpp and avidity dropped as well. Mice immunized with commercial TIV (same route) showed no or very poor IgA response, so no avidity evaluation was performed for this antibody class, in this group. Differences described herein for induced antibody responses (classes and subclasses analyzed) in each intranasally vaccinated group contributed to the observed HAI titers, which were higher for the TIV-IMXQB vaccine (*p* = 0.0006 and *p* = 0.039). This is particularly relevant as HAI titers can be used as a correlate of protection [38].

After the challenge with A/Uruguay/897/2018 (H1N1)dm09-like virus, weight loss was observed in all groups, but mice vaccinated with TIV-IMXQB experienced significantly less weight loss (all mice retained 90% or more of their initial weight) compared to the non-vaccinated group and a more rapid recovery, as evidenced by weight gain from day 6. Recovery of TIV-vaccinated mice took more days, with less than 90% initial weight retention, showing weight increase from day 8. Our group already reported that immunization by intranasal route with *Q. brasiliensis* saponins-based nanoparticle adjuvants induced both mucosal and systemic humoral immunity in adult mice [22,23], and successfully prevented morbidity and mortality caused by influenza virus lethal challenge [23]. Strengthening mucosal immunity due to activation of nasopharyngeal-associated lymphoid tissues (NALT) is pivotal for respiratory viruses such as influenza virus, since studies in mice have shown that intranasal immunization favors local lung-resident B-cell populations that ultimately are responsible of the generation of protective antiviral secretory IgA [37].

Finally, we demonstrated that the passive transfer of a pool of sera from adult mice immunized with the TIV-IMXQB vaccine to experimental infected aged mice promoted a faster recovery, compared to the same practice with a pool of sera from animals immunized with commercial TIV. In fact, passive transfer of sera from mice immunized with commercial TIV or nanoparticle adjuvant only (IMXQB) to infected aged mice did not prevent weight loss associated with viral infection. Furthermore, experimental infected aged mice that received both preparations separately showed a maximum weight loss greater than 10% by day 8 and required many more days to return values close to baseline. Notably, mice that received the pooled sera elicited by TIV-IMXQB began their recovery at day 4 and returned to baseline shortly after.

Previously, we reported that the incorporation of the IMXQB adjuvant to commercial TIV improved the humoral and cellular immunity to TIV antigens, when administered subcutaneously to adult mice [22,23]. In the present study, we also observed that passive transfer of sera from adult mice immunized with TIV-IMXQB to aged mice experimentally infected with influenza virus recovered faster from infection than infected aged mice receiving sera from adult mice immunized with commercial TIV alone. This is highly relevant in the case where passive antibody transfer therapy would be necessary, for example, in vulnerable populations.

## 5. Conclusions

Analyzing the antibody immune response of aged mice immunized twice with commercial TIV with and without *Q. brasiliensis* saponin-based nanoparticles, we demonstrate that the adjuvanted formulation improved the humoral response against the vaccine´s antigens, both qualitatively and quantitatively. Moreover, mice vaccinated with TIV-IMXQB were better able to overcome the challenge of a lethal dose of the A/Uruguay/897/2018 (H1N1)pdm09-like virus, 120 days after the first shot.

We also showed that intranasal administration of a half of a dose of IMXQB-adjuvanted commercial TIV promoted a comparable or even better humoral immune response with the production of antigen-specific IgA antibodies and protected against lethal virus challenge after 120 dpp. Thus, the combination of these two strategies, intranasal delivery and IMXQB nanoparticle platform, represents an attractive approach to improve efficiency of influenza as well as other respiratory virus vaccines for the elderly.

The results reported herein demonstrate that the IMXQB-adjuvanted vaccine conferred a better protection than unadjuvanted vaccines, both by s.c. or i.n. in an aged mice model. These results, together with the fact that incorporating IMXQB nanoparticles into commercial formulation vaccines can be carried out easily, place IMXQB as a very promising adjuvant.

## Figures and Tables

**Figure 1 vaccines-12-00902-f001:**
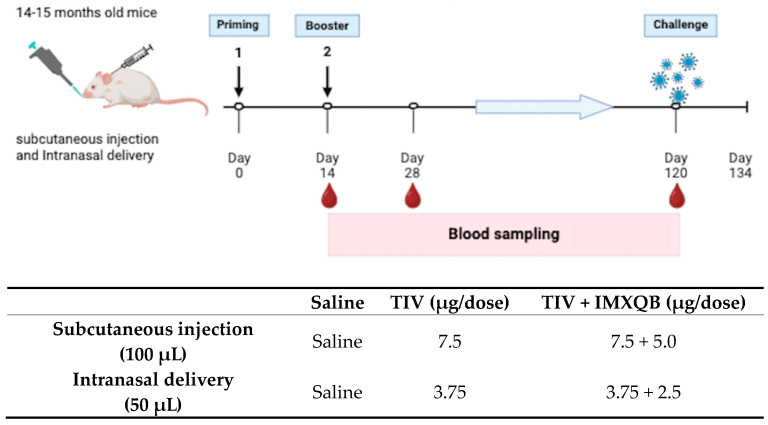
Schematic vaccination, blood sampling, immunogens, and delivery routes of each group. Aged female BALB/c mice were vaccinated by the subcutaneous or intranasal route on days 0 (priming) and 14 (booster) with either TIV-IMXQB or unadjuvanted commercial TIV vaccines. Blood samplings are represented by red drops, and the numbers indicate the days after the prime immunization. For each group, vaccine formulation, dose, and delivery routes are summarized in the table. Created with BioRender.com.

**Figure 2 vaccines-12-00902-f002:**
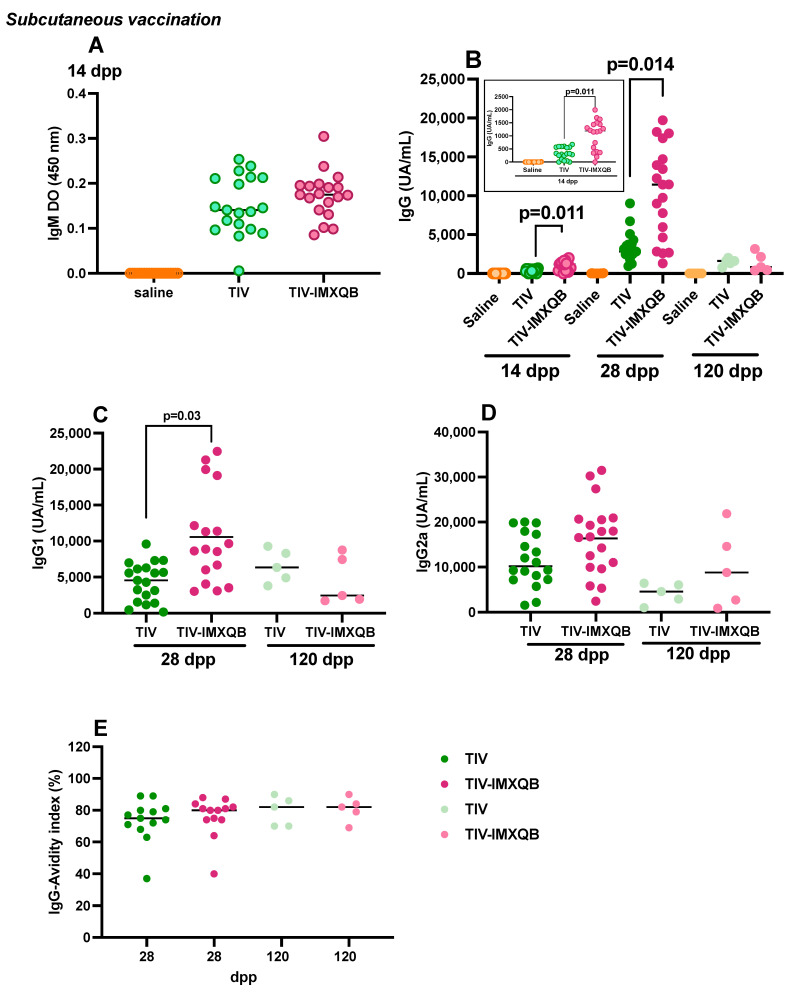
Subcutaneous injections of adjuvanted trivalent influenza vaccine (TIV-IMXQB) elicited higher antibody levels than commercial unadjuvanted vaccine (TIV). Aged female BALB/c mice were vaccinated by the s.c. route on days 0 (priming) and 14 (booster) with either TIV-IMXQB or unadjuvanted vaccine (commercial vaccine). Anti-TIV IgM (**A**) at day 14 and IgG (**B**) at days 14 (*n* = 19 animals per group, empty dots), 28 (*n* = 19 animals per group, strong-colored dots), and 120 (*n* = 5 animals per group, light-colored dots) are represented; the insert shows a zoom view of the IgG values at day 14. IgG1 (**C**) and IgG2a (**D**) antibody levels for 28 and 120 dpp, are shown. Avidity index of IgG antibodies at days 28 and 120 post-priming were assessed (**E**). The median value is indicated by a line, and the dots indicate individual values. The statistical analyses were performed using Kruskal–Wallis and uncorrected Dunn’s post hoc test, comparing the TIV-IMXQB against the TIV group. Statistically significant differences are indicated with the *p* specific value.

**Figure 3 vaccines-12-00902-f003:**
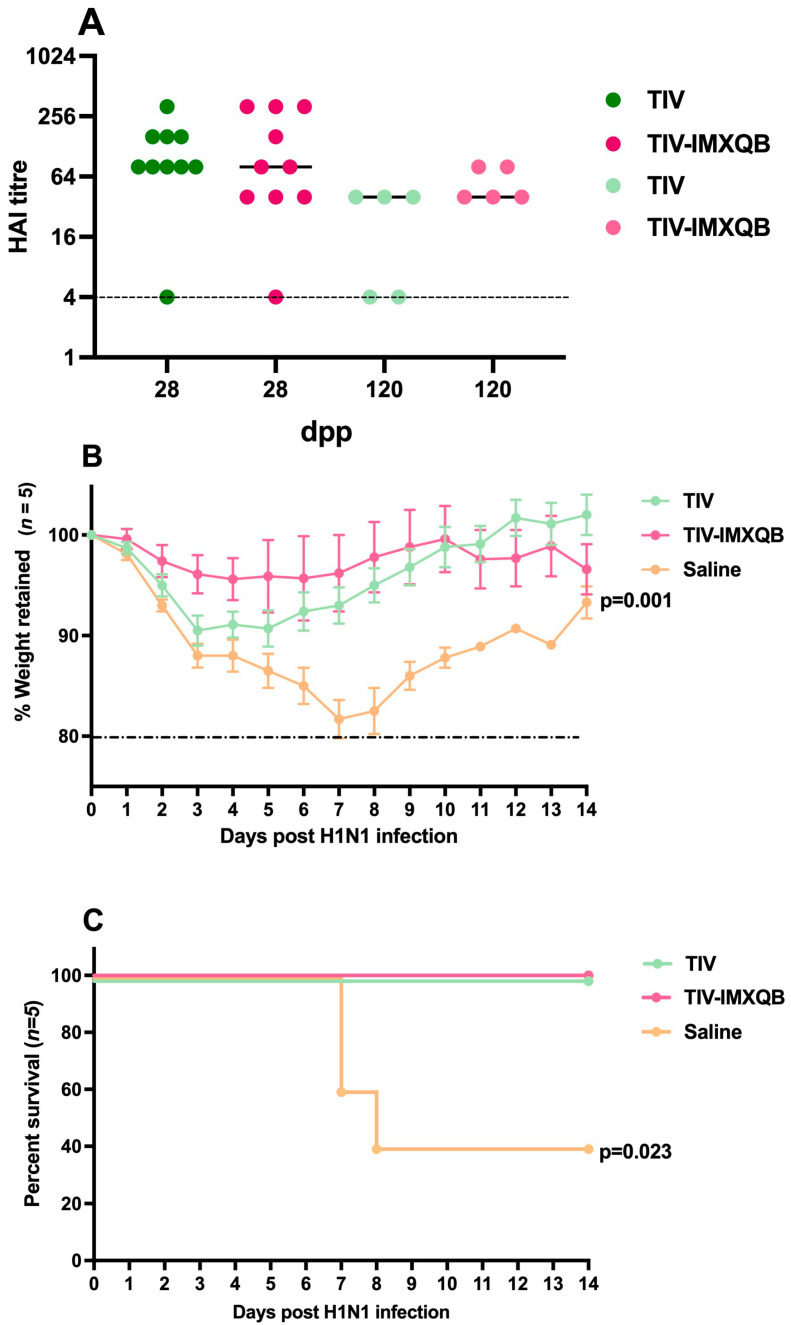
Subcutaneous injections of TIV-IMXQB-adjuvanted vaccine promoted a better protection than TIV. Aged female BALB/c mice. Hemagglutination inhibition antibody titers were measured on days 28 and 120 dpp (*n* = 19 and *n* = 5 animals per group, respectively). A line shows the median mean value, and the dots show individual values. Strong-colored dots: 28 dpp; light-colored dots: 120 dpp (**A**). One hundred twenty dpp, all animals in each group (n = 5 per group, light-colored dots) were intranasally challenged with 1 × 10^6^ TCID (median tissue culture infectious dose)/50 µL of A/Uruguay/897/2018 (H1N1)pdm09 influenza virus and followed during 14 days. Retained body-weight loss (mean and error) (**B**) and percent survival (**C**), are plotted vs. time. The statistical analyses were performed using Mann–Whitney test, comparing TIV-IMXQB against TIV group for each collected serum. Percentage of survival compared to TIV alone was determined by a log-rank (Mantel–Cox) test. Statistical analyses were performed using the non-parametric Kruskal–Wallis test with uncorrected Dunn’s post hoc test for multiple comparisons or a log-rank (Mantel–Cox) test, and each group was compared with the TIV mock group. Significant differences are indicated with the *p* specific value.

**Figure 4 vaccines-12-00902-f004:**
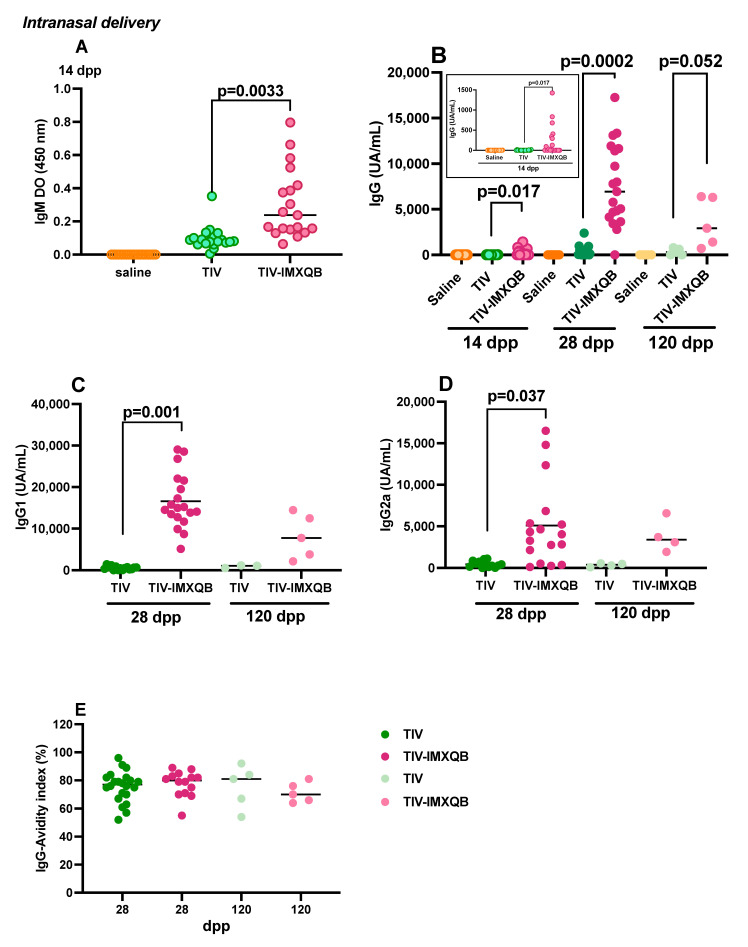
Intranasal delivery of adjuvanted trivalent influenza vaccine (TIV-IMXQB) elicited higher antibody levels than commercial unadjuvanted vaccine (TIV). Aged female BALB/c mice were vaccinated via i.n. on days 0 (priming) and 14 (booster) with either TIV-IMXQB or TIV alone. Anti-TIV IgM (**A**) at day 14 and IgG (**B**) at days 14 (*n* = 19 animals per group, empty dots), 28 (*n* = 19 animals per group, strong-colored dots), and 120 (*n* = 5 animals per group, light-colored dots) are represented; the insert shows a zoom view of the IgG values at day 14. IgG1 (**C**) and IgG2a (**D**) antibody levels for 28 and 120 dpp are shown. Avidity index of IgG antibodies at days 28 and 120 post-priming were assessed (**E**). The median value is indicated by a line, and the dots indicate individual values. The statistical analyses were performed using Kruskal–Wallis and uncorrected Dunn’s post hoc test, comparing the TIV-IMXQB against the TIV group. Statistically significant differences are indicated with the *p*-specific value.

**Figure 5 vaccines-12-00902-f005:**
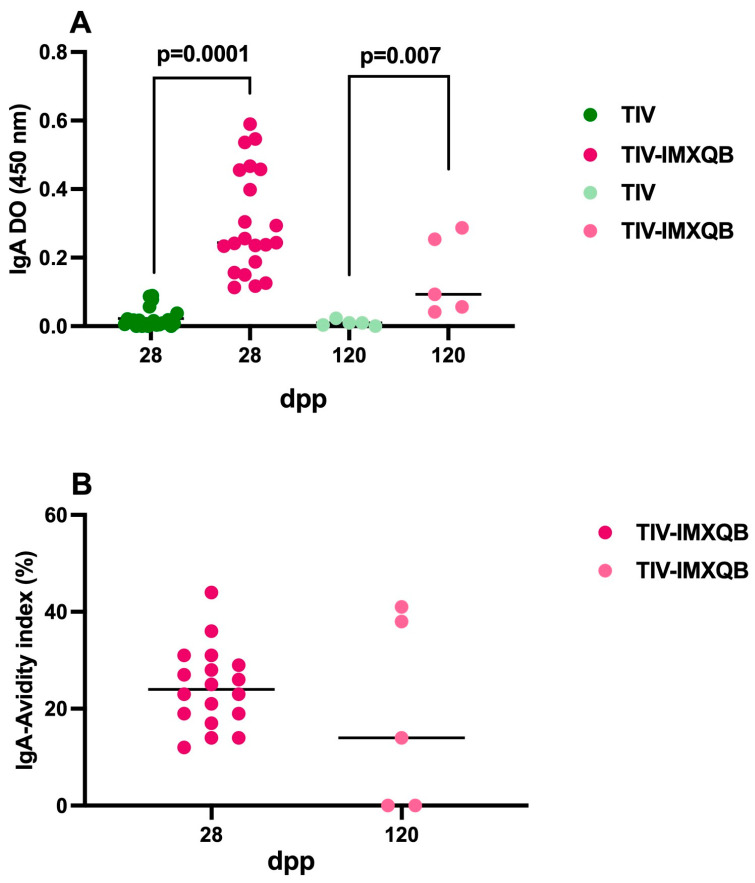
Intranasal delivery of adjuvanted trivalent influenza vaccine (TIV-IMXQB) elicited a significant increase of IgA antibody levels compare to commercial unadjuvanted vaccine (TIV). Anti-TIV IgA were measured on days 28 and 120 dpp (*n* = 19 and *n* = 5 animals per group, respectively). A line shows the median mean value, and the dots show individual values. Strong colored dots: 28 dpp; light colored dots: 120 dpp (**A**). The avidity index of IgA antibodies was evaluated. A line shows the IgA avidity index median value, for de TIV group undetected value, and the dots show individual values (**B**). Statistical analyses were performed using the Mann-Whitney test. Significant differences are indicated with the *p* specific value.

**Figure 6 vaccines-12-00902-f006:**
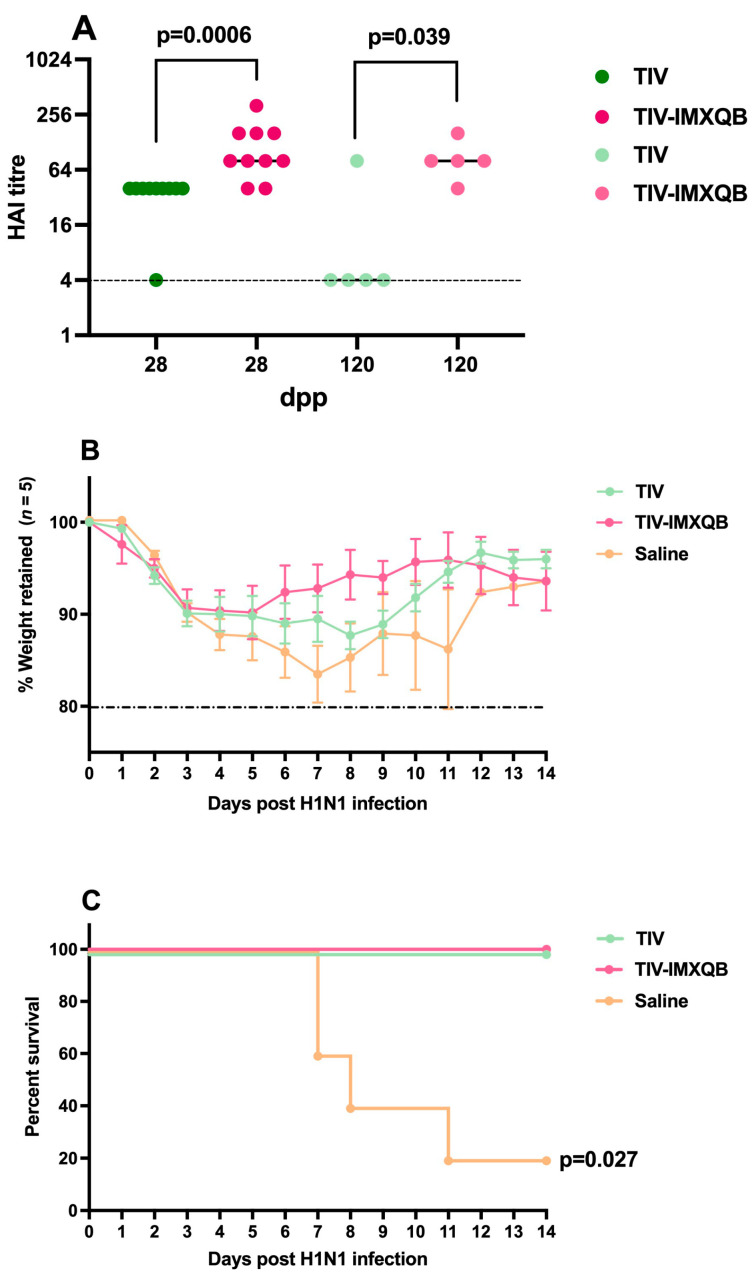
Intranasal delivery of TIV-IMXQB-adjuvanted vaccine induced superior protection than TIV. Hemagglutination inhibition antibody titers were measured on days 28 and 120 dpp (*n* = 19 and *n* = 5 animals per group, respectively). A line shows the median mean value, and the dots show individual values. Strong-colored dots: 28 dpp; light-colored dots: 120 dpp (**A**). One hundred twenty dpp, all animals in each group (n = 5 per group, light-colored dots) were intranasally challenged with 1 × 10^6^ TCID (median tissue culture infectious dose)/50 µL of A/Uruguay/897/2018 (H1N1)pdm09 influenza virus and followed during 14 days. Retained body-weight loss (mean and error) (**B**) and percent survival (**C**), are plotted vs. time. The statistical analyses were performed using Mann–Whitney test, comparing TIV-IMXQB against TIV group for each collected serum. Percentage of survival compared to TIV alone was determined by a log-rank (Mantel–Cox) test. Statistical analyses were performed using the non-parametric Kruskal–Wallis test with uncorrected Dunn’s post hoc test for multiple comparisons or a log-rank (Mantel–Cox) test, and each group was compared with the TIV mock group. Significant differences are indicated with the *p* specific value.

**Figure 7 vaccines-12-00902-f007:**
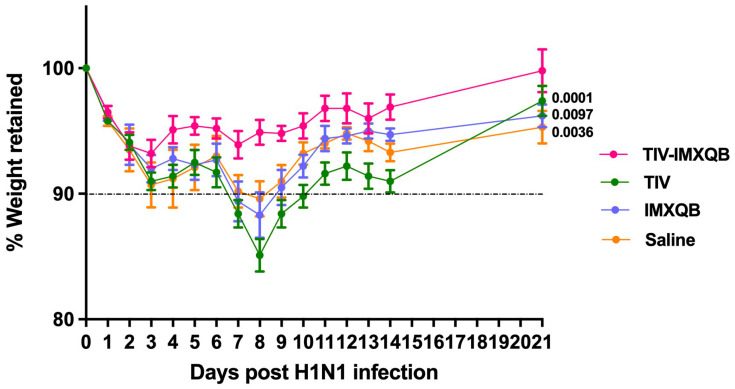
Passive immunization with pooled sera from immunized adult mice with TIV-IMXQB-adjuvanted vaccine contributed to faster recovery of aged mice against sublethal challenge with A/Uruguay/897/2018 (H1N1)pdm09-like virus. Mice aged 15 months old were intranasally infected with a sublethal dose (1 × 10^4^ TCID_50_/50 µL) of A/Uruguay/897/2018 (H1N1)pdm09-like virus. Twenty-four hours after that, the mice received, intraperitoneally, pooled sera from adult animals immunized with TIV (*n* = 7), TIV-IMXQB (*n* = 7), IMXQB (only nanoparticle adjuvant) (*n* = 5), or saline (*n* = 3) and monitored daily for clinical signs of disease, weight loss, and mortality for 21 days. The plot shows body weight retained in each group vs. time (represented by the mean and error). Statistical analyses were performed using the non-parametric Kruskal–Wallis test with uncorrected Dunn’s post hoc test for multiple comparisons, and each group was compared with the TIV-IMXQB group. Significant differences are indicated with the *p*-specific value.

## Data Availability

The data presented in this study are available on request from the corresponding author (FS). e available on request from the corresponding author (FS). orresponding content of this part.

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
