# Peer review of "Intranasal Delivery of Quillaja brasiliensis Saponin-Based Nanoadjuvants Improve Humoral Immune Response of Influenza Vaccine in Aged Mice"

_vaccines, 2024, doi:10.3390/vaccines12080902_

Round 1
Reviewer 1 Report
Comments and Suggestions for Authors
In this manuscript, Intranasal delivery of Quillaja brasiliensis saponin-based nanoadjuvants improve humoral immune response of influenza vaccine in aged mice, the authors reported the subcutaneous or intranasal immunization with TIV-IMXQB induces production of high anti-TIV titers in aged mice, and passive transfer of sera obtained from adult mice immunized with TIV-IMXQB promote a faster recovery of old mice after virus challenge. The results presented in this manuscript are reasonable and interesting. However, the following suggestions may be useful to improve the quality of the manuscript:
1. It is not clear that IMXQB functions as a nanoparticle carrier or an adjuvant, or dual function. This is one of major concerns for the manuscript. The authors stated that “non-adjuvanted TIV vaccine” in the section of “Materials” was used in mouse immunization as an experimental control and compared with TIV-IMXQB. If authors consider IMXQB functions as an adjuvant, thus, it is better to include TIV mixed with an adjuvant in mouse immunization as a study control because adjuvants can enhance a robust immune response.
2. It seems that TIV formulated with nanoadjuvent of IMXQB forms a nanoparticle, which was demonstrated in the publication, “Silveira, F.; Rivera-Patron, M.; Deshpande, N.; Sienra, S.; Checa, J.; Moreno, M.; Chabalgoity, J.A.; Cibulski, S.P.; Baz, M. Quillaja 643 brasiliensis nanoparticle adjuvant formulation improves the efficacy of an inactivated trivalent influenza vaccine in mice. Front. 644 Immunol. 2023, 14, doi:10.3389/fimmu.2023.1163858.”.
3. To weigh the “IMXQB nanoparticles are promising platforms for next-generation viral vaccines.”, it is better to employ other nanoadjuvent, such as LNP, as a control in mouse immunization.
4. The study reported in this manuscript is similar to that described in “Silveira, F.; Rivera-Patron, M.; Deshpande, N.; Sienra, S.; Checa, J.; Moreno, M.; Chabalgoity, J.A.; Cibulski, S.P.; Baz, M. Quillaja 643 brasiliensis nanoparticle adjuvant formulation improves the efficacy of an inactivated trivalent influenza vaccine in mice. Front. 644 Immunol. 2023, 14, doi:10.3389/fimmu.2023.1163858.”. The difference between two studies is the age of mice used in immunizations. Thus, it is better for authors to focus on the comparison of the immunogenic results attained in both reports and illustrate the new findings or insights achieved in this manuscript.
5. The paragraph of “Discussion” is too long to be interesting.
The quality of English language is good, but there are few occurrences such as some grammatical and meaningful expressions confused that require a careful proof prior to publication. For instance, (1) in Figure 4A, unit of Y axis , IgM OD was mistyped as IgM DO; (2) what does it imply in the phrase of “IMXQB nanoparticles are promising platforms for next-generation viral vaccines.”
Comments on the Quality of English LanguageThe quality of English language is good, but there are few occurrences such as some grammatical and meaningful expressions confused that require a careful proof prior to publication. For instance, (1) in Figure 4A, unit of Y axis ,IgM OD was mistyped as IgM DO; (2) what does it imply in the phrase of “IMXQB nanoparticles are promising platforms for next-generation viral vaccines.”
Author Response
Response to Reviewer
General comment from authors: We appreciate and thank the reviewer for the relevant comments and corrections provided, which contribute to significantly improve our manuscript. Changes requested by the reviewer are highlighted in track changes in the manuscript.
Question 1: It is not clear that IMXQB functions as a nanoparticle carrier or an adjuvant, or dual function. This is one of major concerns for the manuscript. The authors stated that “non-adjuvanted TIV vaccine” in the section of “Materials” was used in mouse immunization as an experimental control and compared with TIV-IMXQB. If authors consider IMXQB functions as an adjuvant, thus, it is better to include TIV mixed with an adjuvant in mouse immunization as a study control because adjuvants can enhance a robust immune response.
Authors’ response: Thank you for pointing this out. The IMXQB are made by combining Quillaja brasiliensis saponins and sterols [1]. After that, transmission electron microscopy (TEM) is used to visualize IMXQB nanoparticles, and size distribution is performed using dynamic light scattering (DLS). The combination of saponin fractions with sterols to produce IMXQB in not just a delivery system, even though they have a cage-like structure of about 40 nm diameter in size. Their adjuvant properties rely on several mechanisms not fully understood, as occurs with Matrix-MTM [2]. In previous investigations we demonstrated that the adjuvant effect resulted in an early recruitment of immune cells to the draining lymph node 24 – 48 hours post administration of both QB and IMXQB saponins, pro-inflammatory cytoquines induction [3], and, when combining with immunogens, antigen-specific potent humoral and cellular adaptive immune responses were induced [2,4–7].
Regarding the suggestion of including another experimental group for TIV + another adjuvant, we dismissed it because the main objective of this research was to evaluate whether the commercial formulation of TIV with IMXQB improved the immune response in aged mice, as we previously demonstrated in adults’ mice. In fact, other research groups working with substances with recognized adjuvant properties as CpG, demonstrated an improvement of the immune response in adult mice, but not in aged mice. We have already demonstrated the adjuvant properties of IMXQB in adult mice with different immunogens such as ZIKA virus, bovine viral diarrhea virus and influenza virus, as well as the adjuvant properties of two different saponin fractions from Q. brasiliensis, QB90 and QB80 [4–8].
Our work focuses on comparing TIV with and without adjuvant IMXQB). We have previously used commercial saponins from Quillaja saponaria (IMXQA) as control (see Silveira et al. 2023 [6]). Also, in agreement with the 3R approach on Ethics of Animal Experimentation, which aims to reduce the number of animals, we considered that this control is no longer needed in our researches about IMXQB.
Question 2: seems that TIV formulated with nanoadjuvent of IMXQB forms a nanoparticle, which was demonstrated in the publication, “Silveira, F.; Rivera-Patron, M.; Deshpande, N.; Sienra, S.; Checa, J.; Moreno, M.; Chabalgoity, J.A.; Cibulski, S.P.; Baz, M. Quillaja 643 brasiliensis nanoparticle adjuvant formulation improves the efficacy of an inactivated trivalent influenza vaccine in mice. Front. 644 Immunol. 2023, 14, doi:10.3389/fimmu.2023.1163858.”.
Authors’ response: Please see response in question 1.
Question 3: To weigh the “IMXQB nanoparticles are promising platforms for next-generation viral vaccines.”, it is better to employ other nanoadjuvent, such as LNP, as a control in mouse immunization.
Authors’ response: We concur with this observation. Our group has been working with saponin fractions extracted from native flora, Q. brasiliensis, for more than 15 years. Our aim is to demonstrate that these saponin fractions have adjuvant properties as good as commercial saponins from Q. Saponaria, and its potential use in the formulation of viral vaccines. We do not argue that other adjuvants could be promising for the same purpose as well. We re-phrased our sentence to avoid misunderstanding: “IMXQB nanoparticles may be promising platform for next-generation viral vaccines.”
Question 4: The study reported in this manuscript is similar to that described in “Silveira, F.; Rivera-Patron, M.; Deshpande, N.; Sienra, S.; Checa, J.; Moreno, M.; Chabalgoity, J.A.; Cibulski, S.P.; Baz, M. Quillaja 643 brasiliensis nanoparticle adjuvant formulation improves the efficacy of an inactivated trivalent influenza vaccine in mice. Front. 644 Immunol. 2023, 14, doi:10.3389/fimmu.2023.1163858.”. The difference between two studies is the age of mice used in immunizations. Thus, it is better for authors to focus on the comparison of the immunogenic results attained in both reports and illustrate the new findings or insights achieved in this manuscript.
Authors’ response: In the submitted manuscript, have had commented in the discussion section the results observed in old (this study) and adult mice [6], focusing on the induced immune response from a general point of view, unless otherwise stated. See lines (458 to 477, 506 to 510, 530 to 531, 551 to 555, 570 to 575) in the revised manuscript.
Question 5: The paragraph of “Discussion” is too long to be interesting.
Authors’ response: We believe our discussion is complete as it states previous work from our group as well as a comparison with other studies. Our discussion is in line with our findings.
The quality of English language is good, but there are few occurrences such as some grammatical and meaningful expressions confused that require a careful proof prior to publication.
For instance, (1) in Figure 4A, unit of Y axis , IgM OD was mistyped as IgM DO;
Authors’ response: Thanks for highlighting this. We changed IgM DO for IgM OD in Y axis and revised the whole manuscript as well as the figures.
(2) what does it imply in the phrase of “IMXQB nanoparticles are promising platform for next-generation viral vaccines.”
Authors’ response: Our efforts focus on promoting Q. brasiliensis saponins as a viable alternative to commercial Q. saponaria saponin. In this regard, the saponin-based nanoparticles are also relevant. We have previously established that IMXQB are non-toxic and produce strong immune responses when combined with a variety of viral antigens. Therefore, after our extensive pre-clinical studies performed in the last 15 years, we believe that IMXQB should be evaluated in clinical studies.
References
- Rivera-Patron, M.; Cibulski, S.P.; Miraballes, I.; Silveira, F. Formulation of IMXQB: Nanoparticles Based on Quillaja Brasiliensis Saponins to Be Used as Vaccine Adjuvants. In Methods in molecular biology (Clifton, N.J.); Fett-neto, A.G., Ed.; Methods Mol Biol: Porto Alegre, Brazil, 2022; Vol. 2469, pp. 183–191.
- Stertman, L.; Palm, A.-K.E.; Zarnegar, B.; Carow, B.; Lunderius Andersson, C.; Magnusson, S.E.; Carnrot, C.; Shinde, V.; Smith, G.; Glenn, G.; et al. The Matrix-MTM Adjuvant: A Critical Component of Vaccines for the 21st Century. Hum Vaccin Immunother 2023, doi:10.1080/21645515.2023.2189885.
- Cibulski, S.P.; Rivera-Patron, M.; Mourglia-Ettlin, G.; Casaravilla, C.; Yendo, A.C.A.; Fett-Neto, A.G.; Chabalgoity, J.A.; Moreno, M.; Roehe, P.M.; Silveira, F. Quillaja Brasiliensis Saponin-Based Nanoparticulate Adjuvants Are Capable of Triggering Early Immune Responses. Sci Rep 2018, 8, 13582, doi:10.1038/s41598-018-31995-1.
- Cibulski, S.P.; Mourglia-Ettlin, G.; Teixeira, T.F.; Quirici, L.; Roehe, P.M.; Ferreira, F.; Silveira, F. Novel ISCOMs from Quillaja Brasiliensis Saponins Induce Mucosal and Systemic Antibody Production, T-Cell Responses and Improved Antigen Uptake. Vaccine 2016, 34, 1162–1171, doi:10.1016/j.vaccine.2016.01.029.
- Cibulski, S.; Varela, A.P.M.; Teixeira, T.F.; Cancela, M.P.; Sesterheim, P.; Souza, D.O.; Roehe, P.M.; Silveira, F. Zika Virus Envelope Domain III Recombinant Protein Delivered With Saponin-Based Nanoadjuvant From Quillaja Brasiliensis Enhances Anti-Zika Immune Responses, Including Neutralizing Antibodies and Splenocyte Proliferation. Front Immunol 2021, 12, doi:10.3389/fimmu.2021.632714.
- Silveira, F.; Rivera-Patron, M.; Deshpande, N.; Sienra, S.; Checa, J.; Moreno, M.; Chabalgoity, J.A.; Cibulski, S.P.; Baz, M. Quillaja Brasiliensis Nanoparticle Adjuvant Formulation Improves the Efficacy of an Inactivated Trivalent Influenza Vaccine in Mice. Front Immunol 2023, 14, doi:10.3389/fimmu.2023.1163858.
- Cibulski, S.; Teixeira, T.F.; Varela, A.P.M.; de Lima, M.F.; Casanova, G.; Nascimento, Y.M.; Fechine Tavares, J.; da Silva, M.S.; Sesterheim, P.; Souza, D.O.; et al. IMXQB-80: A Quillaja Brasiliensis Saponin-Based Nanoadjuvant Enhances Zika Virus Specific Immune Responses in Mice. Vaccine 2020, 39, 571–579, doi:10.1016/j.vaccine.2020.12.004.
- Rivera-patron, M.; Baz, M.; Roehe, P.M.; Cibulski, S.P.; Silveira, F. ISCOM-Like Nanoparticles Formulated with Quillaja Brasiliensis Saponins Are Promising Adjuvants for Seasonal Influenza Vaccines. 2021, 1–18, doi:10.3390/ vaccines9111350.
Yours sincerely,
On behalf of the authors,
Fernando Silveira, PhD.
Corresponding author
fsilveira@higiene.edu.uy

Reviewer 2 Report
Comments and Suggestions for Authors
This is considered to be very useful information. The fact that the adjuvants presented here caused more potent immunity, both subcutaneously and by nasal addition, will be very convincing before further human demonstrations. The following are some questions and requests。
L68  Please explain the difference between SBA IMXQB and Matrix-M. Or are they substantially equivalent?
L97 A Google Search on Voucher MVFQ 4321 leads to several papers, I imagine that they probably use the same lot of adjuvant, is this correct? Would changing trees or lots give equivalent results? If the authors have tried it, please write so.
L224 My understanding is that one-way-ANOVA is a parametric method and the Kruskal-Wallis test is a non-parametric method, they are different. If the authors have a reason for choosing a non-parametric method, which tends to be more imprecise, please explain it, even though it probably would not have caused any complications. If they had done the parametric TukeyHSD after ANOVA, this question would not have been asked.
L242 p < 0.01 Please provide specific values, e.g. P=0.005, because P-values are evidence, and so are Figs.2-4.And perhaps more important than the P-value is the fact that the titers are twice as strong compared to the median, which is a surprising effect. Wouldn't it be easier to understand if these values were put in writing? This is also the case for Fig. 5.
Fig.2 If possible, it would be clearer to overlay a violin plot; GraphPad Prism should have such a function. This is also the case for Fig. 3A and Figs. 4 and 5.
L339 Shouldn't this place also show how many times the median value has increased, rather than focusing on the P-value? This is my impression throughout. Yes, the P-value is evidence, but what is more important is how much improvement has been achieved.
Author Response
Response to Reviewer
General comment from authors: We appreciate and thank the reviewer for the relevant comments and corrections provided, which contribute to significantly improve our manuscript. Changes requested by the reviewer are highlighted in track changes in the manuscript.
Question 1: L68. Please explain the difference between SBA IMXQB and Matrix-M. Or are they substantially equivalent?
Authors’ response: Thank you for pointing this out. SBA, is an abbreviation for “Saponin-based adjuvant” and comprises any formulation containing saponins with adjuvant properties, regardless their type, composition, solubility or if they are particulate or not [1,2]. The adjuvant properties rely on several mechanisms not fully understood, as occurs with Matrix-MTM [1] and IMXQB [2].
IMXQB and Matrix-MTM are nanoparticles, similar in shape and size, formed with saponin and sterols, or, saponin, sterols and phospholipids, respectively. They differ in the saponin source. IMXQB are nanoparticles composed by saponin fractions from Quillaja brasiliensis, a native tree mainly from Uruguay, Brazil, Argentina and Paraguay. Matrix-M is composed by saponin fractions from Quillaja saponaria, a native tree from Chile. Matrix-MTM is an adjuvant system from Novavax AB, Uppsala, Sweden and is included in the formulations of some vaccines including COVID-19 [3–5]. As both are cage-like structures, they can be produced, stored and incorporated into the immunogen before injections, in contrast to their predecessors ISCOMs.
In previous investigations we demonstrated an adjuvant effect that resulted in early recruitment of immune cells to draining lymph node 24–48 hours post administration of both QB saponins and IMXQB, pro-inflammatory cytoquines induction [6], and when combining with immunogens, antigen-specific potent humoral and cellular adaptive immune responses were induced [2,7–10].
Over the last decade, our research team has been studying saponins extracted from the leaves of Q. brasiliensis as an alternative to commercial saponins extracted from Q. saponaria tree bark. Comprehensive chemical characterization has indicated that the two species share a plethora of saponin structures, including well-known saponins with significant immunoadjuvant activity, such as QS-21 [11,12].
Question 2: L97 A Google Search on Voucher MVFQ 4321 leads to several papers, I imagine that they probably use the same lot of adjuvant, is this correct? Would changing trees or lots give equivalent results? If the authors have tried it, please write so.
Authors’ response: So far, we've collected leaves from twenty-one trees in Battle Park. Enough leaves were always collected to produce consistent batches of saponins, which were analyzed by TLC [13]. By harvesting a large number of leaves throughout the same season of the year, we ensure a high output of saponin fractions [13,14].
Question 3: L224 My understanding is that one-way-ANOVA is a parametric method and the Kruskal-Wallis test is a non-parametric method, they are different. If the authors have a reason for choosing a non-parametric method, which tends to be more imprecise, please explain it, even though it probably would not have caused any complications. If they had done the parametric TukeyHSD after ANOVA, this question would not have been asked.
Authors’ response: We agree with de reviewer that One-way ANOVA is a parametric method, whereas the Kruskal-Wallis test is a non-parametric method. Given that the data does not follow a normal distribution (they do not meet the assumptions of normality), we performed non-parametric tests.
Question 4: L242 p < 0.01 Please provide specific values, e.g. P=0.005, because P-values are evidence, and so are Figs.2-4. And perhaps more important than the P-value is the fact that the titers are twice as strong compared to the median, which is a surprising effect. Wouldn't it be easier to understand if these values were put in writing? This is also the case for Fig. 5.
Authors’ response: The specific values of p in the graphics and in the manuscript were modified at the reviewer's suggestion.
Question 5: Fig.2 If possible, it would be clearer to overlay a violin plot; GraphPad Prism should have such a function. This is also the case for Fig. 3A and Figs. 4 and 5.
Authors’ response: We thank the reviewer for the graph format recommendation. We recognize the information provided by the violin representation. However, we prefer to show our data with the scatter dot plot as we and the readers can have the exact value from each mouse.
Question 6: L339 Shouldn't this place also show how many times the median value has increased, rather than focusing on the P-value? This is my impression throughout. Yes, the P-value is evidence, but what is more important is how much improvement has been achieved.
Authors’ response: Accepted. Thanks for this suggestion. The median value for TIV and TIV-IMXQB groups were include in the new version of the manuscript (lines 351-354 in the revised manuscript).
References
- Stertman, L.; Palm, A.-K.E.; Zarnegar, B.; Carow, B.; Lunderius Andersson, C.; Magnusson, S.E.; Carnrot, C.; Shinde, V.; Smith, G.; Glenn, G.; et al. The Matrix-MTM Adjuvant: A Critical Component of Vaccines for the 21st Century. Hum Vaccin Immunother 2023, doi:10.1080/21645515.2023.2189885.
- Rivera-Patron, M.; Cibulski, S.P.; Miraballes, I.; Silveira, F. Formulation of IMXQB: Nanoparticles Based on Quillaja Brasiliensis Saponins to Be Used as Vaccine Adjuvants. In Methods in molecular biology (Clifton, N.J.); Fett-neto, A.G., Ed.; Methods Mol Biol: Porto Alegre, Brazil, 2022; Vol. 2469, pp. 183–191.
- Genton, B. R21/Matrix-MTM Malaria Vaccine: A New Tool to Achieve WHO’s Goal to Eliminate Malaria in 30 Countries by 2030? J Travel Med 2023, 30, 1–3, doi:10.1093/JTM/TAAD140.
- Datoo, M.S.; Natama, H.M.; Somé, A.; Bellamy, D.; Traoré, O.; Rouamba, T.; Sorgho, F.; Derra, K.; Rouamba, E.; Ramos-lopez, F.; et al. Articles Efficacy and Immunogenicity of R21 / Matrix-M Vaccine against Clinical Malaria after 2 Years ’ Follow-up in Children in Burkina Faso : A Phase 1 / 2b Randomised Controlled Trial., doi:10.1016/S1473-3099(22)00442-X.
- Dunkle, L.M.; Kotloff, K.L.; Gay, C.L.; Áñez, G.; Adelglass, J.M.; Barrat Hernández, A.Q.; Harper, W.L.; Duncanson, D.M.; McArthur, M.A.; Florescu, D.F.; et al. Efficacy and Safety of NVX-CoV2373 in Adults in the United States and Mexico. New England Journal of Medicine 2022, 386, 531–543, doi:10.1056/nejmoa2116185.
- Cibulski, S.P.; Rivera-Patron, M.; Mourglia-Ettlin, G.; Casaravilla, C.; Yendo, A.C.A.; Fett-Neto, A.G.; Chabalgoity, J.A.; Moreno, M.; Roehe, P.M.; Silveira, F. Quillaja Brasiliensis Saponin-Based Nanoparticulate Adjuvants Are Capable of Triggering Early Immune Responses. Sci Rep 2018, 8, 13582, doi:10.1038/s41598-018-31995-1.
- Cibulski, S.; Teixeira, T.F.; Varela, A.P.M.; de Lima, M.F.; Casanova, G.; Nascimento, Y.M.; Fechine Tavares, J.; da Silva, M.S.; Sesterheim, P.; Souza, D.O.; et al. IMXQB-80: A Quillaja Brasiliensis Saponin-Based Nanoadjuvant Enhances Zika Virus Specific Immune Responses in Mice. Vaccine 2020, 39, 571–579, doi:10.1016/j.vaccine.2020.12.004.
- Cibulski, S.; Varela, A.P.M.; Teixeira, T.F.; Cancela, M.P.; Sesterheim, P.; Souza, D.O.; Roehe, P.M.; Silveira, F. Zika Virus Envelope Domain III Recombinant Protein Delivered With Saponin-Based Nanoadjuvant From Quillaja Brasiliensis Enhances Anti-Zika Immune Responses, Including Neutralizing Antibodies and Splenocyte Proliferation. Front Immunol 2021, 12, doi:10.3389/fimmu.2021.632714.
- Silveira, F.; Rivera-Patron, M.; Deshpande, N.; Sienra, S.; Checa, J.; Moreno, M.; Chabalgoity, J.A.; Cibulski, S.P.; Baz, M. Quillaja Brasiliensis Nanoparticle Adjuvant Formulation Improves the Efficacy of an Inactivated Trivalent Influenza Vaccine in Mice. Front Immunol 2023, 14, doi:10.3389/fimmu.2023.1163858.
- Cibulski, S.P.; Mourglia-Ettlin, G.; Teixeira, T.F.; Quirici, L.; Roehe, P.M.; Ferreira, F.; Silveira, F. Novel ISCOMs from Quillaja Brasiliensis Saponins Induce Mucosal and Systemic Antibody Production, T-Cell Responses and Improved Antigen Uptake. Vaccine 2016, 34, 1162–1171, doi:10.1016/j.vaccine.2016.01.029.
- Cibulski, S.; Rivera-Patron, M.; Suárez, N.; Pirez, M.; Rossi, S.; Yendo, A.C.; de Costa, F.; Gosmann, G.; Fett-Neto, A.; Roehe, P.M.; et al. Leaf Saponins of Quillaja Brasiliensis Enhance Long-Term Specific Immune Responses and Promote Dose-Sparing Effect in BVDV Experimental Vaccines. Vaccine 2018, 36, 55–65, doi:10.1016/j.vaccine.2017.11.030.
- Cibulski, S.; Amorim, T.; Joanda, D.S.; Raimundo, P.; Mangueira, Y.; Silva, L.; Norma, A.; Iris, S.; Paulo, M.; Roehe, M.; et al. ISCOM ‑ Matrices Nanoformulation Using the Raw Aqueous Extract of Quillaja Lancifolia ( Q . Brasiliensis ). Bionanoscience 2022, doi:10.1007/s12668-022-01023-8.
- Yendo, A.; de Costa, F.; Kauffmann, C.; Fleck, J.; Gosmann, G.; Fett-Neto, A. Purification of an Immunoadjuvant Saponin Fraction from Quillaja Brasiliensis Leaves by Reversed-Phase Silica Gel Chromatography. In Methods in Molecular Biology; Fox, C.B., Ed.; Springer New York, 2017; Vol. 1494, pp. 87–93 ISBN 978-1-4939-6443-7.
- Schlotterbeck, T.; Castillo–Ruiz, M.; Cañon–Jones, H.; Martín, R.S. The Use OfLeaves from Young Trees OfQuillaja Saponaria (Molina) Plantations as a New Source of Saponins. Econ Bot 2015, 69, 262–272, doi:10.1007/s12231-015-9320-0.
Yours sincerely,
On behalf of the authors,
Fernando Silveira, PhD.
Corresponding author
fsilveira@higiene.edu.uy
Reviewer 3 Report
Comments and Suggestions for Authors
Dear Editor-in-Chief,
I have now read the manuscript entitled: “Intranasal delivery of Quillaja brasilensis saponin-based nanoadjuvants improve humoral immuneresponse of influenza vaccine in aged mice” by Silveira F et al. (vaccines-3103159). The study is interesting and the authors describe the usefulness of a relatively novel (Q.brasiliensis) saponine-based adjuvants compared with standard influenza trivalent vaccines.
The authors warn that their study population may be a bit small to allow definitive interpretation.
Comments and questions:
Comment 1. The authors present their vaccine administration alternatives used. A consid-erable number of mice receive the vaccine or saline through the nasal cavity and respiratory tract. C1. Still, the study did not investigate presence of mucosal antibodies, why not ?
Comment 2. The administration of vaccine into the nasal cavity in the small Balb/C and CD1 mice used. Interestingly, the authors provide their intranasal/nasal administration volumes of vaccine and control to 50 microliter/mouse. This is a massive amount of fluid flushed into the nasal cavity of a mouse and will not only flood the nasal mucosal tissues but also risk to flush into the lungs and all other respiratory tract tissues !?? This procedure is rather a respiratory tract flush-immunization (as well as an oral/intestinal exposure mucosal immunological tissue vaccine exposure). C2. The authors should comment on the used nasal vaccine volume need and the importance of the chosen volume .
Comment 3. The nasal administration volume would probably represent a volume of vaccine equal to 2 Liters of vaccine delivery through the nose. This would be a quite unwelcome experience of vaccination procedure? C3. The authors could discuss and explain the value and importance of mucosal vaccine volume importance.
Q1. Methods: Paragraphs 2.6 and 2.7. Lines: 194 and 214. The authors present that they challenge mice at 17-18 months of age (Line 194) and in the passive immunization studies (Line 214) 15-months aged mice ? The authors could explain the reason for why differently aged mice were used ? Please, also present how old the life-time of mice can become at their animal facility.
Q2.Paragraph 2.7.Line 202: What was the reason for using CD1 mice instead of BALB/c mice?
Q3a. Methods: (Line 211) Which methods were used to determine neutralization of influenza A and B virus infections in vitro ?
Q3b. The serum IgM, IgA and IgG-titers could be shown for all influenza strains included in the TIV-vaccine and TIV-IMXQB-vaccine immunized animals ?
Q4. Methods: Paragraph 2.2: Experimental vaccines formulations and mice immunizations. (Line 109-120). The authors need to better and clearer explain how they prepare the adjuvant influenza vaccine preparations for the saponin-based adjuvants. The authors claim that they mix appropriate volumes (Line 114) ? Which volumes are appropriate ?
Comments on the Quality of English Language
Dear Editor-in-Chief,
the English language used is sufficient enough to allow presentation and interpretation of the results presented.
Author Response
Response to Reviewer
General comment from authors: General comment from authors: We appreciate and thank the reviewer for the relevant comments and corrections provided, which contribute to significantly improve our manuscript. Changes requested by the reviewer are highlighted in track changes in the manuscript.
Comment 1: The authors present their vaccine administration alternatives used. A consid-erable number of mice receive the vaccine or saline through the nasal cavity and respiratory tract. C1. Still, the study did not investigate presence of mucosal antibodies, why not ?
Authors’ response: We understand the importance of evaluating the presence of antibodies at the mucosal level and we are conscious that represents a study-limiting issue regarding the evaluation of intranasal immunization. We are doing efforts to include it in future investigations. Quantification of secretory IgA (sIgA) in mucosal fluids has several technical difficulties related to compositional variability between samples. It is quite simple to search for the presence or absence of sIgA, however, for comparison between different animals, another parameter is needed to normalize the obtained values and must be chosen carefully to avoid bias; in fact, there is no consensus on the matter. This is not the case for serum samples, so for feasibility reasons, we decided to look for specific IgA at the serum level, as a reflection of the activation of mucosal immunity, since we knew already that when the formulations were administered subcutaneously, this class of immunoglobulin were practically undetectable.
Question 2: The administration of vaccine into the nasal cavity in the small Balb/C and CD1 mice used. Interestingly, the authors provide their intranasal/nasal administration volumes of vaccine and control to 50 microliter/mouse. This is a massive amount of fluid flushed into the nasal cavity of a mouse and will not only flood the nasal mucosal tissues but also risk to flush into the lungs and all other respiratory tract tissues !?? This procedure is rather a respiratory tract flush-immunization (as well as an oral/intestinal exposure mucosal immunological tissue vaccine exposure). C2. The authors should comment on the used nasal vaccine volume need and the importance of the chosen volume.
Authors’ response: The volume of 50 mL/dose is the maximum approved by the ethics committee’s guideline at the University of the Republic (Uruguay) and Université Laval (Canada), institutions to which we belong. Our protocols have always been approved by our ethics committees. A lower volume, example 30 mL/dose could not be used given the protein concentration of the hemagglutinins in the TIV.
Our research team has reported several times intranasal delivery with the maximum allowable volume (50 mL/dose) for mouse challenge [1–3]. In addition, other research groups also use the same volume using the intranasal route [4,5].
Question 3: The nasal administration volume would probably represent a volume of vaccine equal to 2 Liters of vaccine delivery through the nose. This would be a quite unwelcome experience of vaccination procedure? C3. The authors could discuss and explain the value and importance of mucosal vaccine volume importance.
Authors’ response: As noted above, we used the maximum allowable volume for intranasal immunization (50 mL/dose) permitted by our Animal Welfare Ethics Committees. Furthermore, the protein quantity of TIV (90 ug/mL) limited the formulation of the experimental vaccines evaluated in this work.
Question 4: Methods: Paragraphs 2.6 and 2.7. Lines: 194 and 214. The authors present that they challenge mice at 17-18 months of age (Line 194) and in the passive immunization studies (Line 214) 15-months aged mice ? The authors could explain the reason for why differently aged mice were used ? Please, also present how old the life-time of mice can become at their animal facility.
Authors' response: We understand the confusion of the reviewer. We had obtained sera for passive immunization from adult CD1 animals (6-8 weeks old, see line 211 in the manuscript) in 2019, we kept them frozen until use in 2023 for the current study. We did not include this information in the manuscript.
In the current study, aged female BALB/c mice (15-months-old) were infected intranasally and twenty-four hours later, mice were divided into three groups and given pooled sera from TIV immunized CD1 mice (n=7), TIV-IMXQB immunized CD1 mice (n=7), IMXQB immunized CD1 mice (n=5) or saline solution (n=3) (see you line 223-227 in the manuscript).
Today, we can age mice considerably longer. However, in the aging process some animals can die and others can develop tumors (after 15 months). Both events are typical during aging in rodents. For these reasons, we used animals when they reached about 15-17 months.
Question 5: Paragraph 2.7.Line 202: What was the reason for using CD1 mice instead of BALB/c mice?
Authors’ response: This work started before COVID-19 pandemic, but had to be suspended due to the closure of all public facilities. When we reopened, the BALB/c mouse supplier required time to start mouse production. As there was a CD1 colony available at our Institute (Instituto de Higiene), we were able to conduct this experiment with these mice.
Question 6a: Methods: (Line 211) Which methods were used to determine neutralization of influenza A and B virus infections in vitro ?
Authors’ response: Thank you for pointing this out. In the new version of our manuscript (see line 188 to 196 in the revised manuscript), we introduced new text to describe the methodology used to determine the neutralization titers used for the passive immunizations.
Question 6b: The serum IgM, IgA and IgG-titers could be shown for all influenza strains included in the TIV-vaccine and TIV-IMXQB-vaccine immunized animals ?
Authors’ response: We used TIV (the same antigen used to manufacture the vaccine) to sensitize the plates for the in vitro ELISA experiment, and we assessed total antibodies against TIV. However, we challenged with A/Uruguay/897/2018 (H1N1)pdm09-like virus, so we have a direct measure of the anti-H1N1 antibodies produced against this challenge virus.
Question 7: Methods: Paragraph 2.2: Experimental vaccines formulations and mice immunizations. (Line 109-120). The authors need to better and clearer explain how they prepare the adjuvant influenza vaccine preparations for the saponin-based adjuvants. The authors claim that they mix appropriate volumes (Line 114) ? Which volumes are appropriate ?
Authors’ response: Maybe with this change it can be better understood. “TIV-IMXQB was prepared by mixing the required volumes of commercial TIV, IMXQB and saline solutions to obtain 2.5 µg of each HA (TIV) and 5.0 µg of IMXQB in 100 µL/dose. The amount of IMXQB was defined as the saponin content in the nanoparticle adjuvant” (see new line 114 in the revised manuscript).
References
- Rivera-patron, M.; Baz, M.; Roehe, P.M.; Cibulski, S.P.; Silveira, F. ISCOM-Like Nanoparticles Formulated with Quillaja Brasiliensis Saponins Are Promising Adjuvants for Seasonal Influenza Vaccines. 2021, 1–18, doi:10.3390/ vaccines9111350.
- Silveira, F.; Rivera-Patron, M.; Deshpande, N.; Sienra, S.; Checa, J.; Moreno, M.; Chabalgoity, J.A.; Cibulski, S.P.; Baz, M. Quillaja Brasiliensis Nanoparticle Adjuvant Formulation Improves the Efficacy of an Inactivated Trivalent Influenza Vaccine in Mice. Front Immunol 2023, 14, doi:10.3389/fimmu.2023.1163858.
- Cibulski, S.P.; Mourglia-Ettlin, G.; Teixeira, T.F.; Quirici, L.; Roehe, P.M.; Ferreira, F.; Silveira, F. Novel ISCOMs from Quillaja Brasiliensis Saponins Induce Mucosal and Systemic Antibody Production, T-Cell Responses and Improved Antigen Uptake. Vaccine 2016, 34, 1162–1171, doi:10.1016/j.vaccine.2016.01.029.
- Du, Y.; Xu, Y.; Feng, J.; Hu, L.; Zhang, Y.; Zhang, B.; Guo, W.; Mai, R.; Chen, L.; Fang, J.; et al. Intranasal Administration of a Recombinant RBD Vaccine Induced Protective Immunity against SARS-CoV-2 in Mouse. Vaccine 2021, 39, 2280–2287, doi:10.1016/j.vaccine.2021.03.006.
- Ugozzoli, M.; Hagan, D.T.O.; Vaccines, G.S.O.T.T.C.; Ca, E. Intranasal Immunization of Mice with Herpes Simplex Virus Type 2 Recombinant GD2 : The e Ff Ect of Adjuvants on Mucosal and Serum Antibody Responses. 1998.
- Evans, F.; Alí-Ruiz, D.; Rego, N.; Negro-Demontel, M.L.; Lago, N.; Cawen, F.A.; Pannunzio, B.; Sanchez-Molina, P.; Reyes, L.; Paolino, A.; et al. CD300f Immune Receptor Contributes to Healthy Aging by Regulating Inflammaging, Metabolism, and Cognitive Decline. Cell Rep 2023, 42, doi:10.1016/j.celrep.2023.113269.
Yours sincerely,
On behalf of the authors,
Fernando Silveira, PhD.
Corresponding author
fsilveira@higiene.edu.uy

Round 2
Reviewer 1 Report
Comments and Suggestions for Authors
Thank authors for their clear responses. Hope the readers can perceive that “the main objective of this research was to evaluate whether the commercial formulation of TIV with IMXQB improved the immune response in aged mice, as we previously demonstrated in adults’ mice.”